# A Numerical Investigation of Enhancing Microfluidic Heterogeneous Immunoassay on Bipolar Electrodes Driven by Induced-Charge Electroosmosis in Rotating Electric Fields

**DOI:** 10.3390/mi11080739

**Published:** 2020-07-30

**Authors:** Zhenyou Ge, Hui Yan, Weiyu Liu, Chunlei Song, Rui Xue, Yukun Ren

**Affiliations:** 1School of Mechatronics Engineering, Harbin Institute of Technology, West Da-zhi Street 92, Harbin 150001, China; gezhenyou1997@163.com (Z.G.); sclei@hit.edu.cn (C.S.); 15145011764@163.com (R.X.); rykhit@hit.edu.cn (Y.R.); 2School of Electronics and Control Engineering, Chang’an University, Middle-Section of Nan’er Huan Road, Xi’an 710064, China; 3State Key Laboratory of Robotics and System, Harbin Institute of Technology, West Da-Zhi Street 92, Harbin 150001, China

**Keywords:** on-chip immunoassay, induced-charge electroosmosis, rotating electric field, bipolar floating electrode, microfluidics

## Abstract

A unique approach is proposed to boost on-chip immuno-sensors, for instance, immunoassays, wherein an antibody immobilized on the walls of a microfluidic channel binds specifically to an antigen suspended freely within a working fluid. The performance of these sensors can be limited in both susceptibility and response speed by the slow diffusive mass transfer of the analyte to the binding surface. Under appropriate conditions, the binding reaction of these heterogeneous immuno-assays may be enhanced by electroconvective stirring driven by external AC electric fields to accelerate the translating motion of antigens towards immobilized antibodies. To be specific, the phenomenon of induced-charge electroosmosis in a rotating electric field (ROT-ICEO) is fully utilized to stir analyte in the vicinity of the functionalized surface of an ideally polarizable floating electrode in all directions inside a tri-dimensional space. ROT-ICEO appears as a consequence of the action of a circularly-polarized traveling wave signal on its own induced rotary Debye screening charge within a bipolar induced double layer formed on the central floating electrode, and thereby the pertinent electrokinetic streamlines exhibit a radially converging pattern that greatly facilitates the convective transport of receptor towards the ligand. Numerical simulations indicate that ROT-ICEO can enhance the antigen–antibody binding reaction more effectively than convectional nonlinear electroosmosis driven by standing wave AC signals. The effectiveness of ROT-ICEO micro-stirring is strongly dependent on the Damkohler number as well as the Peclet number if the antigens are carried by a continuous base flow. Our results provide a promising way for achieving a highly efficient heterogeneous immunoassay in modern micro-total-analytical systems.

## 1. Introduction

Immunoassays, which is known as the specific binding reaction between the free antigens suspended in a fluid flow and the immobilized antibodies at a functionalized surface, have been broadly applied in medical diagnostics, quality control, biomarker identification, and environment monitoring for their high selectivity [1,2]. Traditional immunoassays, including microarrays and enzyme-linked immunosorbent assay (ELISA), usually require a vast sample volume and long cultivation time due to the complicated fluid handling steps involved at different stages of the assay [3,4,5,6]. Some representative label-free biomolecule detection approaches have been applied to probe specific proteins and bacteria [7,8]. Nevertheless, the dependence of the test performance on the interfacial binding reaction, which is routinely limited by slow diffusion rate and long detection time, limits their broad application in various situations [9,10]. In addition, the low throughput constrains the application of immunoassay in time-pivotal circumstances [11]. With the rapid development of microfabrication techniques, embedding the functional biosensors into micro-total-analytical systems (μTAS) has captured great attention from the microfluidic society [12]. In stark contrast with the conventional immunoassay approaches, microfluidic biosensors provide enormous advantages to transport the antigen sample onto the binding surface in a continuous-flow mode, in terms of imposing a quite low demand on the sample volume and generating high-throughput while with a much shorter analysis time and an enhanced sensitivity [13]. The mass transfer process where the specific analyte is conveyed from the bulk of the liquid suspension to the reaction surface is negatively influenced by the molecular diffusion effect across a concentration gradient on account of a small Reynolds number in microsystems [13]. Consequently, on-chip immunoassays still suffer from restrictions in both response speed and detection sensitivity, due to the slow diffusive transport of target antigens to the functionalized surface with immobilized antibodies, which determines a non-ideal device performance in diffusion-limited binding reactions with fast surface reactions [14].

A lot of physical mechanisms have been applied to result in stirring flow patterns in microfluidic channels for on-chip biomedical diagnosis, such as hydrodynamic pressure [15], electrokinetics [16,17,18,19], magnetic effect [20], and optical forces [21]. For instance, Selmi et al. calculated the binding kinetics using a microchannel-based flow confinement strategy with an orthogonal complementary stream to guide the translating motion of analyte molecules towards the binding surface [22]. Alternatively, AC electrokinetic (ACEK) phenomenon has become a popular way to drive fluid motion and manipulate colloid samples within to expected locations in the presence of a low voltage supply [23,24,25,26]. ACEK mainly includes dielectrophoresis (both particles and stratified liquid contents) [27], AC electrothermal induced flow (ACET) [28,29,30], AC electroosmosis (ACEO) [31,32,33], and induced-charge electroosmosis (ICEO) [34,35,36]. Both ACET and ACEO have been theoretically and experimentally exploited for accelerating the convective transport of antigens towards specific antibodies for enhancing the binding rate on functionalized electrode surfaces [37,38]. On the other hand, ICEO has emerged as a brand new tool for manipulation of both fluid and particle motion in microfluidic channels [39]. Like ACEO, ICEO is originated by the action of the applied electric field on its own induced charge within a thin induced double layer (IDL) on a polarizable solid surface immersed in an electrolyte solution and has been broadly applied for pumping [40], mixing [41,42,43], and particle handing [44,45]. Though ICEO has a similar mechanism with ACEO, the introduction of floating conductors (namely, floating electrodes) endows ICEO the traits of flexible configuration, locally addressable, free from external wiring, and easy for device integration as compared to ACEO. Our group have recently employed ICEO vortex flow field to trap and enrich microscale particle samples in both static and dynamic conditions with a low voltage supply [46,47,48,49]. Pascall et al. designed a standard electrode structure for actuating ICEO slipping fluid motion on the ideally polarizable surface of a floating metal strip in the center of the gap between a pair of conducting probes inserted into the reservoirs on both sides, and demonstrated by strict experimental measurement and physical argumentation that the presence of a dielectric coating layer and an ion adsorption effect on the central floating electrode (FE) was responsible for the larger ICEO slipping velocity predicted by the standard RC circuit theory than that from experiments [50].

As a result, when the reaction rate is essentially much quicker than the diffusive transport of target analytes to the sensor surface in microchannels, it is necessary to explore a label-free and highly integrated detection strategy to boost the binding rate between antigen and antibody in diffusion-limited cases. To address this issue, in this study, we propose an approach to transport the target antigen sample to the binding surface in a dilute electrolyte (typically with electric conductivity less than 0.05 S/m, so as to evade double-layer shrinkage and steric effect at higher ion concentrations) and improve the device performance of microfluidic heterogeneous immunoassays by using ICEO electroconvective streaming on ideally polarizable surfaces of FE. In particular, a traveling-wave voltage signal with a 90° phase shift is imposed sequentially to a circular electrode array of four discrete phases. ICEO fluid motion with a radially converging flow pattern is induced on the central FE by such a rotating electric field, namely, the phenomenon of ROT-ICEO. By numerical modeling, ROT-ICEO is shown to be more effective in delivering the antigen samples in all directions and suppressing the length of the diffusion boundary layer than ordinary nonlinear electroosmosis driven by AC standing wave signals, due to the action of its tri-dimensional chaotic streamlines.

## 2. Theory and Methods

### 2.1. Device Geometry 

To investigate the vortex flow pattern of ROT-ICEO and its important role in accelerating the binding rate in an immuno-transducer, a microfluidic chip with a coplanar array of thin-film microelectrode is designed and displayed in Figure 1. A square conducting FE with a width of *W_F_* is deposited on the bottom surface of a microfluidic chamber with a height of *H_C_* and a length of *L_C_*, respectively. A circular array of 4 rectangular driving electrodes (DE) is disposed on the channel bottom surface, and surrounds the central FE. By imposing a 90°-phase-shifted travelling wave (TW) voltage signal to the 4 DE in sequence, a rotating (ROT) electric field is produced and runs throughout the chamber. Since the TW signal rotates circularly, instead of propagating in a linear route, the resulted electric field is in effect a circularly polarized rotating (ROT) electric field. It is noteworthy that the rotating field rotates counterclockwise in the direction of the decreasing voltage phase, so we call it a counterclockwise rotating electric field. Figure 2 is an advanced device design under dynamic incoming laminar streams, which will be introduced in detail in Section 2.3.

The external ROT field injects bipolar counterionic charges into a thin Debye layer at the FE/electrolyte interface by Ohmic conduction, in the presence of a normal field component on the blocking electrode at the early stage. After a characteristic RC time scale $\tau_{\mathrm{RC}}=RC_{D}/\sigma_{f}\left(1+\delta \right)$ for double-layer capacitive charging, the bulk electric field lines are fully repelled, and a stable bipolar induced double layer (IDL) is developed on the ideally polarizable surface of the FE in the field center due to complete field-induced Debye screening as shown in Figure 3a. From the perspective of an observer, this renders the conducting surface manifest as a perfect insulator beyond a characteristic distance scale of the Debye screening length, giving rise to a pair of ICEO micro-vortices in opposite rotating directions on top of the central FE due to the interaction of the tangential field components with the dipolar induced ionic charges inside the bipolar IDL (Figure 3a). The above physical picture is valid only under a steady DC bias. For a low-frequency rotating electric field, however, the situation becomes more subtle. As the field vector rotates counterclockwise within each complete AC voltage cycle, with the axis of rotation constantly fixed at the field center, the bipolar charge accumulated within the IDL rotates synchronously in the direction of the electric field as well. In this way, the ICEO slipping fluid motion switches alternatively between two complementary convection modes, in which the flow direction is along the *x*- and *y*-axis, respectively (Figure 3). After time average operation, the ROT-ICEO fluidic eddies ought to exhibit a convergent profile, with the liquid molecules sucked from surrounding medium in all outer directions to the center of the conducting surface of the FE and then ejected to the bulk fluid along the channel depth direction, as indicated by the 3D toroidal arrows in Figure 1. This kind of electroconvective stirring flows on top of the bipolar FE may be in favor of enhancing antigen-antibody specific binding reactions on the functionalized interface of the FE.

### 2.2. Basic Theory of ICEO Electroconvection at the Metal/Electrolyte Interface

To account for the occurrence of transient ICEO streaming on metal electrodes with analytical convenience, complex notation is invoked for various electric field variables. For instance, ϕ(t)=Acos(ωt+θ)=Re(Aejωtejθ)=Re(ϕ˜ejωt), where $\tilde{\phi }$ is the complex amplitude of the transient AC voltage ϕ(t), and Re() the real part operator. *A*, *ω* and *θ* denote the amplitude, angular field frequency and phase angle of the imposed AC voltage signal, respectively. Under sinusoidal steady-state, the charge conservation equation within the liquid domain is reduced to the Laplace equation:(1)$$\nabla^{2}\tilde{\phi }=0$$

Under the Debye–Huckel limit, the IDL behaves like a thin capacitor skin being charged by the conduction current from the resistance of liquid bulk. Consequently, the Ohmic current in the bulk should be continuous with the displacement current running across the thin boundary layer at the electrode/electrolyte interface:(2)$$\sigma n\cdot \nabla \tilde{\phi }=j\omega C_{0}\left(\tilde{\phi }-{\tilde{\phi }}_{0}\right)$$
where ***n*** denotes the unit outward vector normal to the thin-film electrode, pointing from the reaction surface to the bulk of the liquid suspension, and σ the electrolyte conductivity. $C_{0}=C_{D}/\left(1+\delta \right)$ is the total double-layer capacity, which is essentially a series connection of the diffuse layer capacitance $C_{D}=\varepsilon /\lambda_{D}$ and Stern layer capacitance *Cs* = 0.8 F/m^2^, in terms of the surface capacitance ratio $\delta =C_{D}/C_{S}$. $\tilde{\phi }$ and ${\tilde{\phi }}_{0}$ are the electrostatic potential in the bulk fluid right outside the IDL and the equal potential of the metal electrode, respectively.

The peripheral DE array located adjacent to the channel sidewalls is powered by a TW voltage signal, with the specific voltage sequence $\phi_{0}^{1}=$
*A*cos(*ωt*), $\phi_{0}^{2}=$
*A*cos(*ωt* + 90°), $\phi_{0}^{3}=$
*A*cos(*ωt* + 180°), $\phi_{0}^{4}=$
*A*cos(*ωt* + 270°) imposed to the four DE strips along the clockwise direction, while the central square electrode is floating in potential, as shown in Figure 1. 

Only the voltage drop across the diffuse layer $\tilde{\zeta }={\tilde{\phi }}_{0}-\tilde{\phi }/1+\delta$ serves as the effective induced zeta potential (IZP) that contributes to the induced electrokinetic flows. The time-averaged ICEO slipping under AC forcing is derivable from the generalized Helmholtz–Smoluchowski formula, which is subsequently inserted into the Navier–Stokes equation as a leaking wall boundary condition on the ideally polarizable surfaces of all the metal electrodes:(3)〈uslip〉=−εη12Re(ζ˜E˜t*)=ε2η11+δRe((ϕ˜−ϕ˜0)(E˜−E˜·n·n)*)
where η is dynamic viscosity of water, <…> the time-average operator for calculating the averaged value within one sinusoidal voltage cycle, and the asterisk * the complex conjugate.

Fluid flows within the microchannel obeys the simplified Navier–Stokes equation for water-based incompressible Newtonian fluids:(4)$$-\nabla p+\eta \nabla^{2}u=0$$
(5)∇·u=0
where *p* denotes the hydrostatic pressure, and ***u*** the vector field of flow velocity originated by combined ICEO and external pressure gradient. It is well known that ICEO fluid motion vanishes in high conductivity buffer solutions due to both double layer shrinkage and ion overcrowding phenomenon inside the IDL. So, we prefer to study herein the effect of ROT-ICEO on improving microfluidic immunoassays in dilute electrolyte with electric conductivity usually no more than 0.02 S/m. The aqueous electrolyte is a typical Newtonian fluid, and has a constant dynamic viscosity, serving as the most appropriate liquid medium for suspending the free antigens in the present analysis. On the other hand, recently, the necessity of manipulation of biofluids in small confinements has triggered a renewed interest in the dynamics of non-Newtonian fluid with a shear-rate-dependent viscosity. However, this is beyond the scope of current work with the purpose to provide a utilitarian reference for the selection of the parametric space in experiments with water-based Newtonian fluid. As a result, a uniform viscosity value *η* = 0.001 Pa·s is used in the simulation to reconstruct the actual mechanical behavior of target antigens monodispersed in water solution. Please refer to Ref. [51,52,53] for a systematic knowledge of both Newtonian fluid and non-Newtonian fluid.

### 2.3. Mass Transfer of Antigen and Binding Reaction Enhancement

Mass conservation of the target antigens freely suspended in the liquid flow can be mathematically described by the standard convection-diffusion equation:(6)$$\frac{\partial c}{\partial t}+\nabla \cdot J=0$$
(7)J=uc−D∇c
where *c* stands for the concentration of antigens in the bulk fluid, and *D* their mass diffusivity. An initial background concentration of antigens is introduced to the microchamber in the static case (Figure 1) or to the channel inlet in the dynamic condition (Figure 2).

The specific binding reaction between the antigen in liquid suspension and the antibodies adhered to the surface of the FE can be presumed to abide by the first-order Langmuir adsorption model:(8)$$\frac{\partial B}{\partial t}=k_{\mathrm{on}}C_{w}(R_{T}-B)-k_{\mathrm{off}}B$$
where *B* represents the surface concentration of the antigen bound on the reaction surface in mol/m^2^, k_on_ and k_off_ denote the association rate constant and the dissociation counterpart, respectively. *Cw* is the antigen volumetric concentration just on top of the reaction surface, and *R_T_* the surface concentration of the fixed antibody in mol/m^2^. 

To enable the quantification of the feasibility of ROT-ICEO micro-stirring in elevation of the antigen-antibody binding rate, we define the transient binding enhancement factor, *Be*(*t*) = *B_V_*(*t*)/*B*_0_(*t*), in which *B_V_*(*t*) and *B_0_*(*t*) are the bound antigen concentration after introducing ROT-ICEO slipping flow and without voltage supply at time node *t*, respectively. It has been reported that the binding rate is also highly dependent on the nondimensional Damkohler number:*Da* = k_on_*R_T_H_C_*/*D*(9)
which is the ratio of the reaction speed to the diffusion speed. The *Da* number is usually employed to judge whether the biosensor is restricted by diffusion or by reaction. If the reaction rate is quicker than the diffusion transport of target analyte to the sensor surface, the whole binding process is diffusion-limited. On the contrary, the binding rate will be reaction-limited in the situation whereby the analyte diffusion is fast but the reaction speed cannot match with the rate of diffusion.

## 3. Results and Discussion

### 3.1. Binding Reaction Enhancement by ROT-ICEO Micro-Stirring

Three distinct convection modes of ICEO in this four-phase rotating electrode array are compared in detail in the Appendix A, with the corresponding powering schemes explicitly presented in Table 1. As the optimum ICEO slipping flow profile is created by case (i), we then focus on the effect of ROT-ICEO micro-stirring on the antigen-antibody binding reaction on the functionalized surface of the central FE. Before the DE array is energized, in the absence of ICEO micro-stirring, the concentration of antigen freely suspended in the fluid bulk is primarily depleted by the molecular diffusion effect. The mass transfer limitation constrains severely the binding reaction between the free antigen and the fixed antibody, and results in the rapid growth of the antigen-depleted diffusion boundary layer (Figure 4a), the thickness of which determines the detection performance of the proposed immunosensor. When the four metal strips in the peripheral DE array are excited by a TW voltage sequence with neighboring electrodes of a 90° phase shift at voltage amplitude *V*_0_ = 8 V and field frequency *f* = 200 Hz, the transversal ROT-ICEO recirculating vortex flow field stirs effectively the fluidic sample and thereby redistributes the depleted antigen concentration on top of the central square FE. As displayed in Figure 4b, the rotating electrokinetic whirlpools enhance the convection effect on the functionalized surface of the FE, facilitating greatly the efficient transport of free antigens in the bulk to the reaction surface. Consequently, the depletion boundary layer shrinks in thickness, and depletion occurs only in the central region of the FE surface (Figure 4b), which provides a sufficient chance for a binding interaction between the free antigen and immobilized antibody, leading to an enhancement of both the association and disassociation rates.

To explore the dependence of the bound antigen concentration on the applied voltage magnitude, we calculated by numerical simulation the binding rate in the presence of ROT-ICEO micro-stirring under sinusoidal steady-state, with respect to the non-improved situation with no voltage supply, when *Da* number equals 660. As identified in Figure 5a, when the four DE strips are activated with varying voltage amplitude of 0 (the passive case), 2, 4, 8, 16, and 32 V with a prescribed signal frequency of 200 Hz (the active case), the resulted binding rate becomes monotonously higher with larger imposed voltages. In the low voltage range (0–8 V), the binding rate is almost linearly proportional to the voltage amplitude, in that the enhanced ROT-ICEO whirlpool promotes the convective transport of free antigen to the functionalized surface for diffusion-limited binding reactions. In accordance with the classical RC-circuit theory of ICEO, our numerical modeling indicates that the ROT-ICEO slipping velocity has a quadratic dependence on the imposed AC voltage amplitude, and increases by 4-fold as the voltage is doubled (Figure 5b). As the voltage further increases beyond 8 V, the growth trend of the binding rate slows down, since the reaction speed on the functionalized surface cannot match the fast ROT-ICEO electroconvection, namely the binding reaction transits from being diffusion-limited to being reaction limited as the time-averaged ROT-ICEO slipping velocity on the ideally polarizable surface under harmonic AC forcing exceeds a certain threshold value. In this sense, there always exists an optimal AC voltage for the improvement of the specific binding reaction.

### 3.2. Effect of the Damkohler Number

To investigate the importance of ROT-ICEO rotating whirlpool in microfluidic immunoassays, we calculate the binding enhancement factor *Be* of the binding reaction on the application of an AC signal with *V*_0_ = 4 V and *f* = 200 Hz for a time period of *t* = 100 s. As shown in Figure 6a, the antigen-antibody binding rate enhances globally with an increase of the Damkohler number, implying a higher binding efficacy yielding a factor of 21.5 higher binding rate for an applied TW voltage of 4 V at 200 Hz with a *Da* number of 1000. For a small *Da* number, ICEO vortex flow on top of the FE is not able to enhance the binding efficiency by transporting electro-convectively the free antigens to the binding surface due to the slow reaction confinement. On the other hand, when *Da* is sufficiently large and surpasses a certain threshold, the reaction rate becomes too fast, and therefore not enough time is left for association. For instance, once the DE array is excited by 4 V at 200 Hz, the ROT-ICEO micro-stirring is not potent enough, so that the binding performance reaches a plateau when *Da* number is beyond 10,000 (Figure 6a). Under this situation, the voltage amplitude has to be further elevated, so as to get a larger ICEO flow velocity for convective delivery of free antigens towards the functionalized surface and lead to an obvious improvement in the binding rate. On the basis of the above analysis, a higher *Da* number (*Da* = 1000) is chosen for subsequent simulations.

Considering the ROT-ICEO circulating fluid motion has a second-order dependence on the voltage magnitude imposed on the electrode array, we then study the relationship between the binding rate enhancement and the AC voltage amplitude when the Da number is fixed at *Da* = 1000. As indicated in Figure 6b, when an AC signal at an intermediate field frequency *f* = 200 Hz is applied to the peripheral DE array, the interfacial binding rate rises quickly by adjusting the source voltage magnitude and is almost linearly proportional to the background electric field intensity with the increment of the voltage amplitude below 16 V (Figure 6b). The reason behind is that, since the reaction rate on the conducting surface of the central FE is fast, the diffusive transport nature of the free antigen determines there is no possibility for efficient binding reaction to occur. In the presence of ROT-ICEO micro-vortices, however, electroconvection accelerates the mass transfer of antigens to the functionalized surface and enhances the binding performance. 

With further increase of the applied voltage, the binding rate enhances to a slight degree as the voltage exceeds 16 V, which can be explained by the excessively large ROT-ICEO slipping velocity in the lateral direction on the ideally polarizable surface of FE in contrast to the limited reaction rate. In fact, the larger applied voltage is certainly important for the acceleration of the binding reaction, while the actual binding rate may be negatively influenced by a sufficiently large voltage because the reaction rate is not able to match with the moving velocity of the free antigens to the binding surface any longer. That is, the antigen-antibody binding reaction turns from being diffusion-limited to being reaction limited at large voltages. So, it will be meaningless to simply increase the voltage amplitude for boosting the binding response of the microfluidic immunosensors. It is then necessary for us to seek other possible ways to achieve the same goal.

### 3.3. Frequency-Dependent Binding Reaction

The ideal operating condition of the immunosensor is supposed to depend on the signal frequency, in that the ROT-ICEO slipping velocity itself is very susceptible to the field frequency of the applied TW voltage (Appendix A). As shown in Figure 7b, the ROT-ICEO fluid motion attains a localized relaxation peak at an intermediate frequency between the inverse RC time constant for electrochemical polarization of the peripheral DE array and that of the central FE. Namely, $f_{RC}^{DE}$ ≤ *f_ideal_* ≤ $f_{RC}^{FE}$, which agrees well with the preceding analysis in Section 3.1. According to the simulation result in Figure 7b, the ideal working frequency of ICEO is *f_ideal_* = 200 Hz, any further deviation of the field frequency from this critical value would make the ICEO flow velocity decay to a great extent whatever the applied voltage is 4 V, 8 V, or 16 V. For instance, the slipping fluid motion decreases by almost 50% as the signal frequency increases from the ideal frequency of 200 Hz to a higher value of 1000 Hz (Figure 7b). Although the binding enhancement factor Be is also maximized at the same ideal frequency 200 Hz, it decreases no more than 50% with the increment in frequency from 200 Hz to 1000 Hz (Figure 7a), which can be ascribed to the fact that not only the ROT-ICEO slipping fluid motion but also the molecular diffusion effect contribute to the immuno-reaction. In practical experiments, it is quite suitable for us to raise the field frequency to 1000 Hz or even 2000–10,000 Hz (Figure 7a), since the binding rate decays much more slowly than the ROT-ICEO slipping fluid motion itself as the field frequency increases (Figure 7b). By doing this, we can insist on improving the binding performance of the microfluidic immuno-sensors without having to know the accurate ideal operating frequency in advance and evade the bipolar electrode reaction and bubble production on top of the central FE that tend to occur below 1000 Hz at the same time.

### 3.4. Effect of Geometric Arrangement of the Floating Electrodes

In this section, we study how the geometric configuration of the floating electrodes in the field center exerts an influence on the binding efficiency enhancement at a given TW voltage amplitude of *V*_0_ = 4 V. In preceding discussions, the square FE in the center of the rotating electric field has a given edge length of *W_F_* = 100 μm, being comparable in size with the channel depth *H_C_* = 200 μm. So, the first choice for changing the geometry is to adjust the edge length of the individual FE while maintaining its quantity. 

As shown in Figure 8b, the ideal working frequency shifts to lower values as the FE’s edge length rises from 100 μm to 300 μm, keeping consistent with the theoretical prediction of $f_{ideal}=\sigma \lambda_{D}\left(1+\delta \right)/2\pi \varepsilon R$, in which an increment of the characteristic macroscopic length scale *R* for electrochemical polarization of the FE results in a smaller RC charge relaxation frequency at the FE/electrolyte interface. It is well known that ICEO flow velocity becomes faster as the size of the FE increases, as evidenced by the simulation result of ROT-ICEO slipping fluid motion for different FE edge lengths (Figure 8b) as well.

As for the corresponding enhancement of binding performance, however, a marked difference has emerged (Figure 8a). Although the ideal operating frequency for the microfluidic immunosensor shifts to lower frequency values just like the ideal ICEO flow rate, the maximum binding enhancement factor Be decreases as the FE becomes larger in size (Figure 8a). A plausible reason for this particular difference is that the ROT-ICEO slipping velocity is indeed fastest on the surface of the largest FE (the blue line in Figure 8b), but the ICEO vortex flow field in the bulk fluid tends to be suppressed to a great degree by a vertical confinement effect as the size of FE approaches and even exceeds the channel depth (Figure 8c). Consequently, the recirculating fluid motion in the liquid bulk due to the action of ROT-ICEO cannot be fully developed in the presence of a finite channel height (Figure 8c), which may exert an adverse effect on the binding reaction enhancement. This conclusion suggested that it is not necessary to deliberately enlarge the size of the square FE for accelerating the analyte mass transfer towards the functionalized surface, and there exists an intermediate edge size on the order of the channel depth serving as the best choice for such purpose.

The second way to reconfigure the geometry of the central FE is to alter the number of discrete floating electrodes located in the center of the microchamber. As shown in Figure 9c, we introduce a 3 × 3 coplanar FE array to take the place of the original individual FE surrounded by the four peripheral DE metal strips. In the absence of AC power, the target analytes suspended in the bulk are consumed by the diffusion-limited interfacial binding reaction, resulting in the formation of one thick depleted boundary layer on the ideally polarizable surfaces of each of the 9 floating conductors respectively, as shown in Figure 9c. On switching the multiphase function generator on, ROT-ICEO whirlpools appear on all the FEs, which are constantly sucked from the surrounding medium to the electrode center, then ejected upward to the top surface of the fluidic chamber, and finally spread out in all directions to form closed fluid loops. As a consequence, all the depletion boundary layers are reshaped by ROT-ICEO vortex flow field, becoming much thinner on the electrode surface with most of the depletion region shifted above the center of the floating conductors (Figure 9d), which indicates an enhancement in electroconvective transport of the free antigens to the functionalized surfaces, bringing about more opportunities for the binding reaction between the immobilized antibody and target analyte.

Even so, the binding rate is different from one another on the 9 FE. The binding enhancement factor *Be* is the least for the middlemost FE (*Be* = 16 for the 5th electrode at *f* = 200 Hz), and a bit larger on the outer ones (*Be* ranges from 18 to 25 for the 1st–4th and 6th–9th electrodes at *f* = 200 Hz). This unexpected growth phenomenon of the bound antigen concentration on the peripheral eight electrodes can be accounted for by the difference in ICEO slipping velocity within the microfluidic system (Figure 9b). As shown in Figure 9b, the ICEO slipping fluid motion is weakest on the middlemost FE as well, corresponding to the least improvement of the binding efficiency (Figure 9a). The unequal electrokinetic flow velocity on top of the 9 FE can be ascribed to the following two reasons: (1) The discrete arrangement of the coplanar FE array makes the electric field unevenly distributed in the fluidic chamber, which is stronger on the outer eight FEs and weakest on the middlemost FE. Since the electroosmotic flow velocity is linearly proportional to the intensity of the tangential electric field component that reaches a local minimum in the field center, the nonlinear ROT-ICEO slipping velocity is suppressed to a certain degree on the 5th FE. (2) The ACEO fluid motion has a net flow component that propagates in the direction of the rotating electric field (Appendix A), which can be effectively superimposed with the ICEO vortex flow field on the outer eight electrodes, but diminishes gradually along the field gradient due to the action of finite penetration depth, resulting in a higher electroosmotic flow rate on the surface of the 8 FE close to the DE array than that on the middlemost one that is most far away from chamber sidewalls. The above two factors coact to make the ideal *Be* number *Be* = 16 (the purple line in Figure 9a) on the functionalized surface of the central FE lower than *Be* = 22 (the black line in Figure 8a) with merely a single FE. However, the largest *Be* = 27 on the 1st FE in the 3 × 3 discrete FE arrangement is still much higher than *Be* = 22 in the 1 × 1 configuration. The large variance of bound antigen concentration on different FEs in the advanced device design (Figure 9a,d) may be useful under certain circumstances that desire a range of binding rates simultaneously at distinct positions of the inhomogeneous immunoassay.

### 3.5. Binding Reaction Enhancement in a Pressure-Driven Flow

To deal with a more realistic situation in a microfluidic device with continuous sample injection, we have to check the effectiveness of the radially converging ROT-ICEO slipping fluid motion on top of the FE in a straight microchannel under the influence of an axial pressure gradient on the antigen-antibody binding reaction. For such, we prescribed the boundary condition at the upstream channel entrance (Figure 10) with a paraboloidal profile of a mean inlet flow rate *u*_0_ = 25–200 μm/s in the full scale tri-dimensional computational domain. In this integrated device design, two straight branch channels on both sides are bridged by a microchamber in the center, as shown in Figure 10. In contrast to the static condition, there is a uniform concentration of the analyte in the solution at the initial time, and it is assumed that all the analyte sample is brought in by the incoming laminar flow, so that a fixed analyte concentration *c* = *c*_0_ is prescribed at the channel inlet, while no analyte is present in the bulk for the initial condition.

As shown in Figure 10a, without AC power, the axial laminar flow driven by the externally-imposed pressure difference constantly injects the free antigens into the microchannel, and the base flow is paraboloidal in essence (Figure 11a). So, the supplement of reacted analyte adjacent to the binding surface in the center of the chamber bottom surface mainly depends on the axial convective transport of the laminar streaming and transverse mass diffusion. On the application of a phase-shifted TW signal to the four peripheral DE metal strips, however, the rotating ROT-ICEO whirlpool appears (Figure 11b) and severely distorts the free antigen concentration on top of the central FE, providing more chance for binding interaction between the ligand and receptor in terms of a much thinner depletion boundary layer on the FE surface (Figure 10b). Consequently, the transient motion of target analytes is a consequence of the combined action of laminar inflow, molecular diffusion effect, and ICEO micro-stirring under the dynamic condition. In particular, the vortex flow field of ROT-ICEO is of the greatest importance in delivering the suspended antigens transversely to the functionalized reaction surface and refreshing the depleted sample as signified by the non-negligible analyte concentration gradient above the FE surface (Figure 10b), giving rise to enhanced binding reaction kinetics.

Since there is no background analyte concentration in the bulk before the incoming electrolyte carries the sample into the channel, it takes a finite time for the biosensor to make a definite response to the analyte flow. The specific time duration equals the ratio of the distance (from the channel entrance to the flat FE) to the inlet flow rate. So, a lower pump speed implies a larger response time for the binding reaction (Figure 12a). For instance, with a low inlet flow rate of *u*_0_ = 25 μm/s, the surface concentration of bound antigen begins to increase observably only after *t* = 100 s. On the contrary, as the pump velocity is raised to *u*_0_ = 200 μm/s, the relaxation time needed for the binding reaction to make a response to the applied voltage decreases sharply from *t* = 100 s to *t* = 10 s.

Under the circumstances with a net inlet flow rate, the ratio of convection to diffusion rate, namely, the Peclet number *Pe* = *uH*/*D*, has to be introduced to depict the correlation between the binding kinetics and inlet flow rate. We calculate the binding enhancement factor *Be* as a function of *Da* number for distinct Peclet number, namely, 400, 1000, 2000, and 4000, which correspond to a pump speed of 20 μm/s, 50 μm/s, 100 μm/s, and 200 μm/s, respectively (Figure 12b). As shown in Figure 12b, the value of *Be* enhances as the *Da* number increases under a given inlet flow rate, which agrees well with the previous discussions on the *Da*-dependence of the sensor performance in static condition (Figure 6a). The influence of *Pe* number on the binding reaction is much more pronounced than that of *Da* number (Figure 12b). With an increasing *Pe* number, the value of *Be* first rises when *Pe* is no more than 1000, and then declines as *Pe* further increases from 1000 to 4000. This non-monotonous varying trend of *Be* as a function of *Pe* number is attributed to the following two reasons: (1) Since *Be* is calculated at a specific time node of *t* = 300 s, the binding reaction cannot be well activated by the imposed AC voltage as the samples are not sufficiently transported to the functionalized surface for a small *Pe* (*Pe* ≤ 1000). (2) In the presence of a large enough pump speed (*Pe* > 1000), the free antigens can amply arrive at the top of the binding surface. Although a higher pump speed is able to boost the analyte transportation along the channel length direction, a finite ICEO vortex flow field in the lateral direction is less efficient for higher inlet flow rates by causing a lower probability in the binding reaction between the immobilized antibody and the dynamic antigens that pass too quickly over the functionalized surface. So, in a practical experiment, the microfluidic immunoassay should operate at a moderate Peclet number or inlet flow velocity to achieve an observable binding rate enhancement, and either a lower or higher pump rate may deteriorate the device functionality in biosensing.

## 4. Conclusions

In summary, a unique ROT-ICEO method is proposed herein for the enhancement of the specific binding reaction between free antigens and immobilized antibody on a functionalized surface of conducting floating electrodes deposited in the center of the channel bottom surface for performing microfluidic heterogeneous immunoassays. The lateral ROT-ICEO whirlpools can create more chances for the binding reaction by convectively transporting the free antigens to the transducer surface, resulting in an acceleration of both the association and dissociation processes. ROT-ICEO in a circularly polarized electric field favors a slipping flow pattern whereby the electroosmotic streamlines converge radially from all outer directions to the electrode center, and is thereby more advantageous to achieve full-scaled micro-stirring in comparison with the conventional ICEO slipping modes. The dependence of binding efficiency enhancement on AC voltage amplitude, signal frequency, and discrete arrangement of the floating electrodes with or without an external pump fluid motion is analyzed in detail. It is believed that the detection susceptibility and limit of detection are both enhanced by implementing the ROT-ICEO vortex flow field on the ideally polarizable surface of the blocking FE. The numerical computations reported here demonstrate that ROT-ICEO micro-stirring can serve as a utilitarian technique for boosting the binding rate in microfluidic heterogeneous assays, especially when the reaction process is dominated by a slow mass diffusion.

## Figures and Tables

**Figure 1 micromachines-11-00739-f001:**
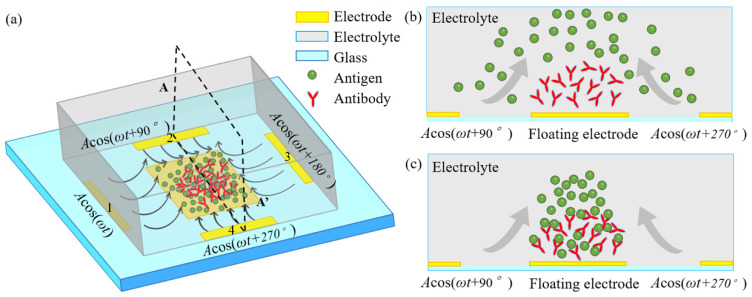
A systematic schematic of improving the specific binding reaction between immobilized antibodies and freely suspended antigens in the dilute electrolyte by induced-charge electroosmosis driven by an externally-imposed rotating electric field (ROT-ICEO) on the ideally polarizable surface of a central floating electrode (FE). (**a**) A 3D sketch for enhancing inhomogeneous immunoassay in a static microchamber; (**b**) fluidic samples before electrical powering; (**c**) electroconvective transport of antigens towards the functionalized FE surface accelerates the interfacial binding reaction.

**Figure 2 micromachines-11-00739-f002:**
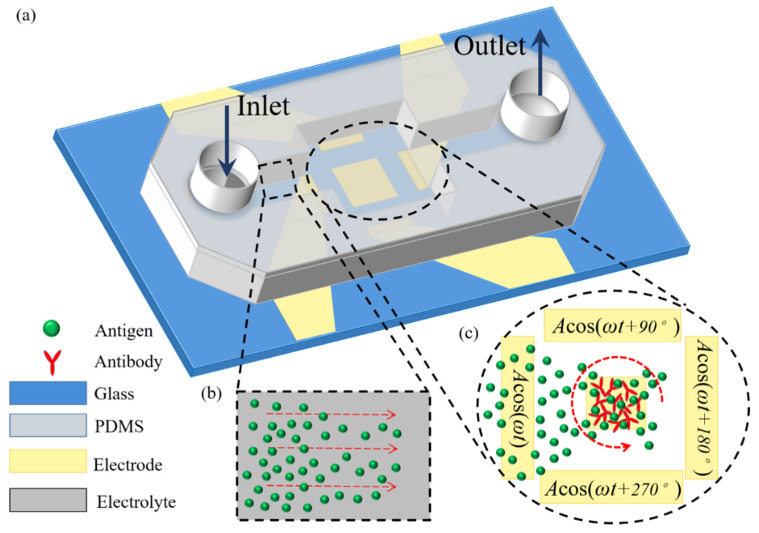
A 3D schematic illustration of ROT-ICEO-mediated antigen-antibody specific binding reaction on the conducting surface of the central FE encircled by a four-phase DE strip array, under the dynamic condition that an external pressure-driven laminar flow injects the target antigen samples from the inlet to the downstream outlet. Once they pass on top of the central FE, the localized ROT-ICEO vortex flow field disturbs the laminar streamlines and transports the antigens to the FE’s ideally polarizable surface, giving rise to more chance for the binding reaction between antigens and antibodies. (**a**) A 3D schematic of the dynamic immuno-sensor in the presence of continuous base flow. (**b**) A magnified view of the upstream flow passage before the suspended antigens pass by the reaction region. (**c**) An amplified view of the critical reaction region on top of the electrode array affected by electroconvection of ROT-ICEO.

**Figure 3 micromachines-11-00739-f003:**
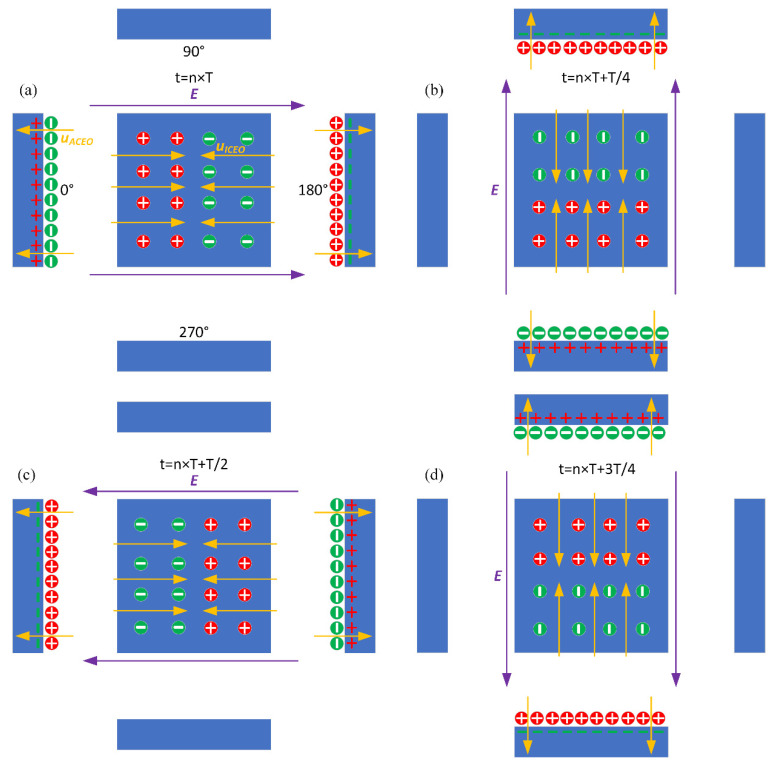
Theoretically-predicted bi-dimensional phase diagram of the applied rotating electric field lines, induced surface charge, and the resulted transient ROT-ICEO surface slipping flow at different time instants within a complete AC cycle for case (i): (**a**) *t* = *nT*, (**b**) *t* = *nT* + *T*/4, (**c**) *t* = *nT* + *T*/2, and (**d**) *t* = *nT* + 3*T*/4.

**Figure 4 micromachines-11-00739-f004:**
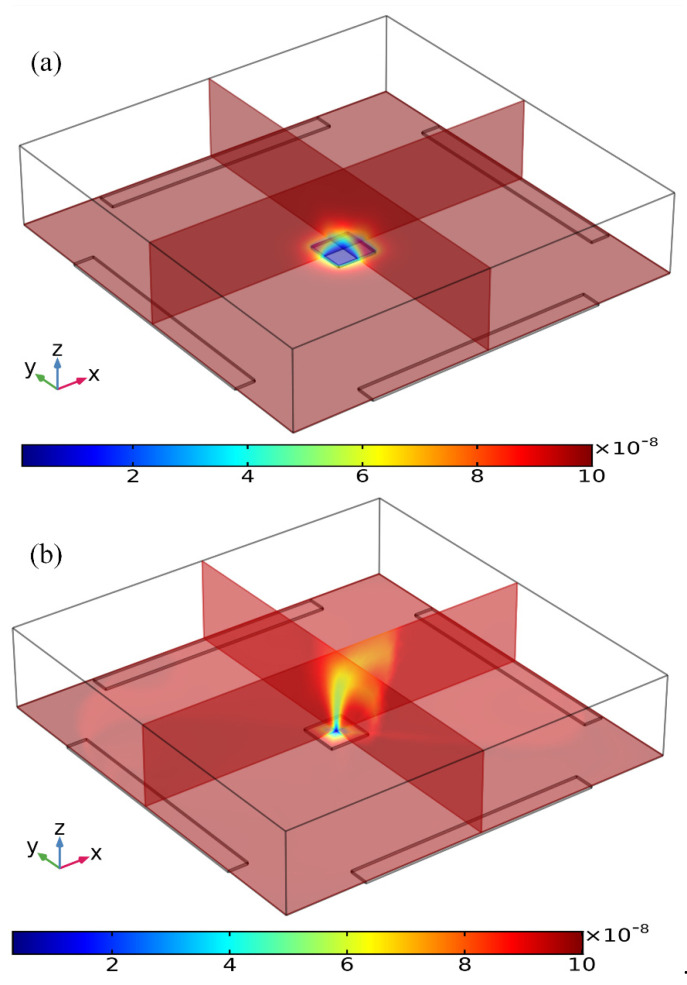
Simulation result of the volumetric concentration of suspended antigens on top of the binding surface at *t* = 100 s for the following two circumstances: (**a**) Concentration distribution of suspended antigens in the absence of any AC voltage supply with only analyte diffusive transport on the FE, leading to a thick depletion boundary layer. (**b**) The ROT-ICEO recirculating flow stirs effectively the depleted concentration after the peripheral DE array is powered by a four-phase AC voltage signal at *V*_0_ = 8 V and *f* = 200 Hz.

**Figure 5 micromachines-11-00739-f005:**
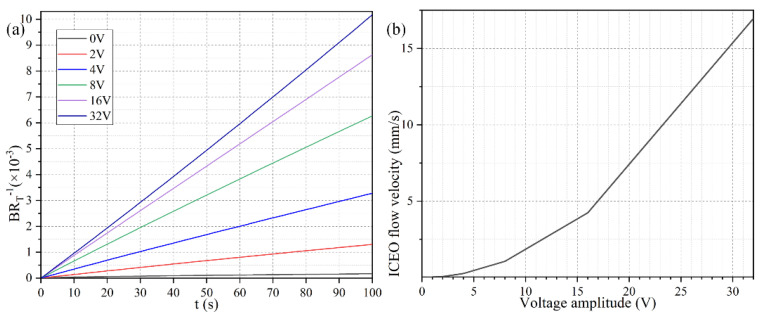
(**a**) Normalized concentration evolution of the bound antigen with respect to the immobilized antibody on the functionalized surface of FE affected by ROT-ICEO under varying AC voltage at *f* = 200 Hz. (**b**) ROT-ICEO slipping velocity as a function of the applied voltage at *f* = 200 Hz.

**Figure 6 micromachines-11-00739-f006:**
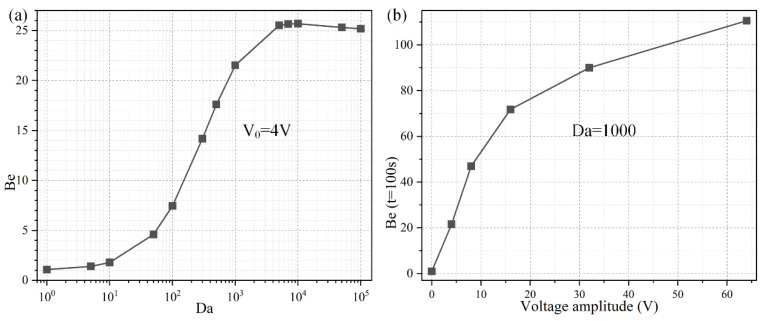
Effect of Da number on the binding efficiency improvement due to the action of ROT-ICEO slipping flow. (**a**) A data point plot of the binding enhancement factor Be with respect to an increasing Da number under AC voltage *V*_0_ = 4V and field frequency *f* = 200 Hz. (**b**) Voltage-dependent Be for different voltage amplitudes at *t* = 100 s when *Da* = 1000.

**Figure 7 micromachines-11-00739-f007:**
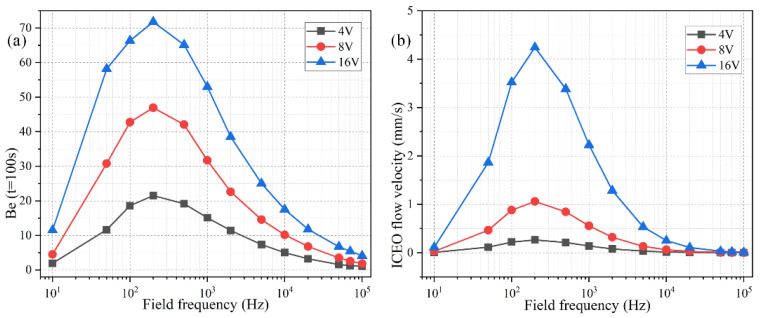
Effect of the AC signal frequency on the binding reaction with *Da* = 1000. (**a**) A data point plot of the binding enhancement factor *Be* versus AC signals of varying oscillation frequencies when the voltage amplitude takes the value of *V*_0_ = 4 V, 8 V, and 16 V, respectively. (**b**) ROT-ICEO slipping velocity as a function of the field frequency for distinct voltage amplitude.

**Figure 8 micromachines-11-00739-f008:**
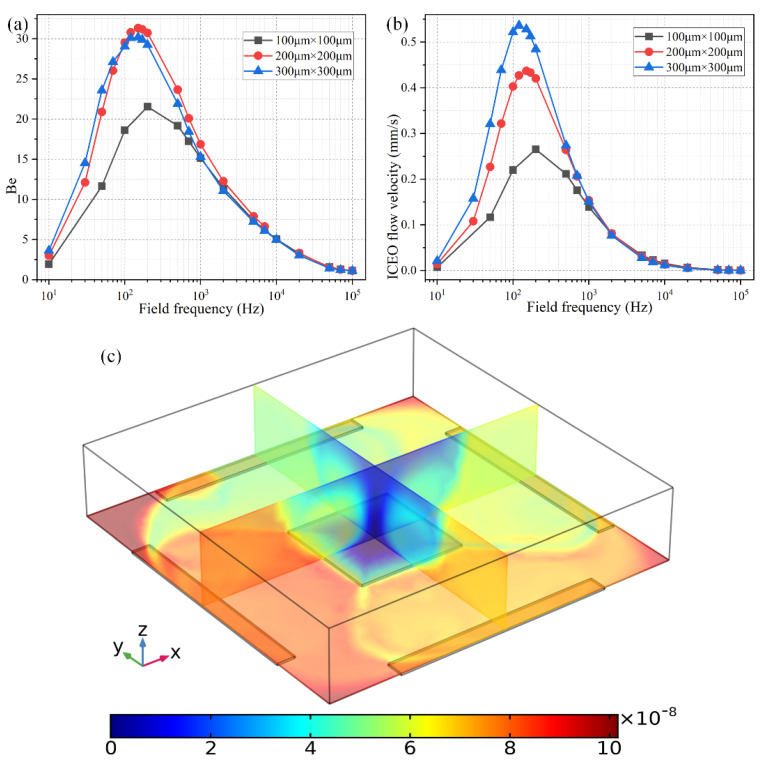
Effect of the FE’s size on the binding reaction improvement at a given voltage amplitude *V*_0_ = 4 V. (**a**) Frequency-dependent *Be* for different FE’s edge length; (**b**) Frequency-dependent ROT-ICEO slipping velocity when the edge length of the central FE takes the value of 100 μm, 200 μm, and 300 μm, respectively. (**c**) A cross-sectional plot of the suspended antigen concentration on top of a 300 μm × 300 μm FE with an AC voltage supply of *V*_0_ = 4 V and *f* = 150 Hz at *t* = 100 s.

**Figure 9 micromachines-11-00739-f009:**
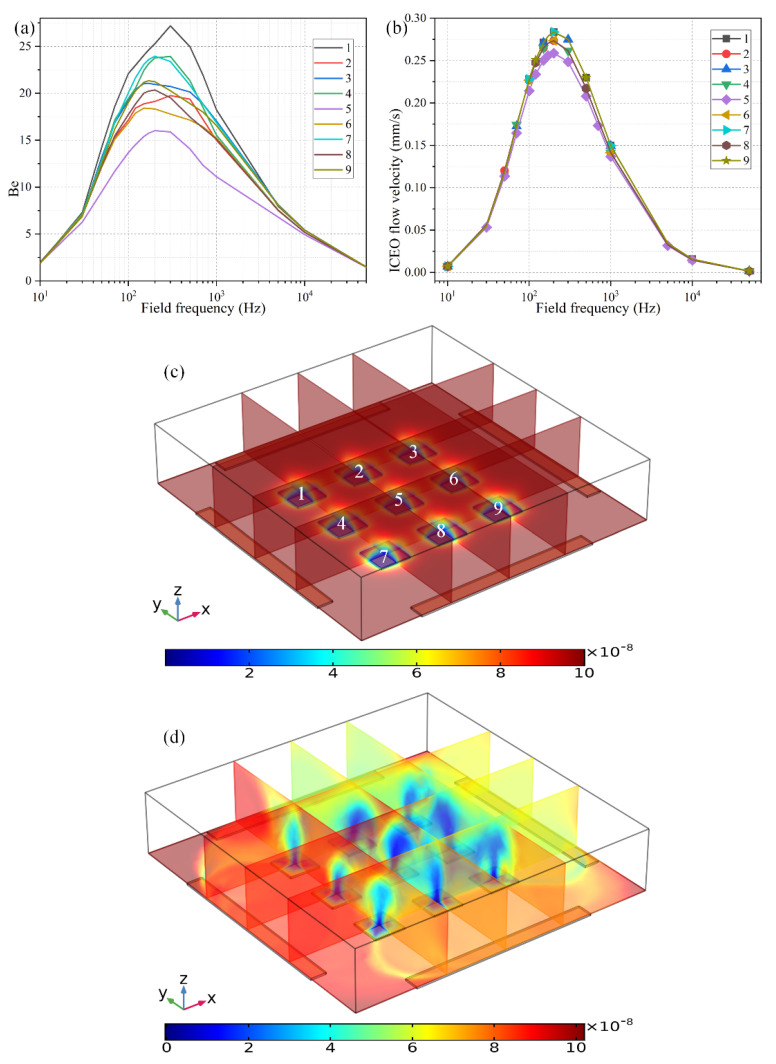
The effect of the number of FE in the discrete electrode arrangement on the degree of binding reaction improvement when the voltage amplitude of the applied rotating electric fields is *V*_0_ = 4V. (**a**) The binding enhancement factor *Be* on each electrode in the FE array versus AC signals of different field frequencies at *t* = 100 s. (**b**) Frequency-dependent ROT-ICEO slipping velocity on each electrode in the FE array. (**c**) Multiple cross-sectional plots of the suspended antigen concentration with no external voltage supply at *t* = 100 s. (**d**) Multiple cross-sectional plots of the suspended antigen concentration at *t* = 100 s with an imposed rotating electric field at *V*_0_ = 4 V and *f* = 200 Hz.

**Figure 10 micromachines-11-00739-f010:**
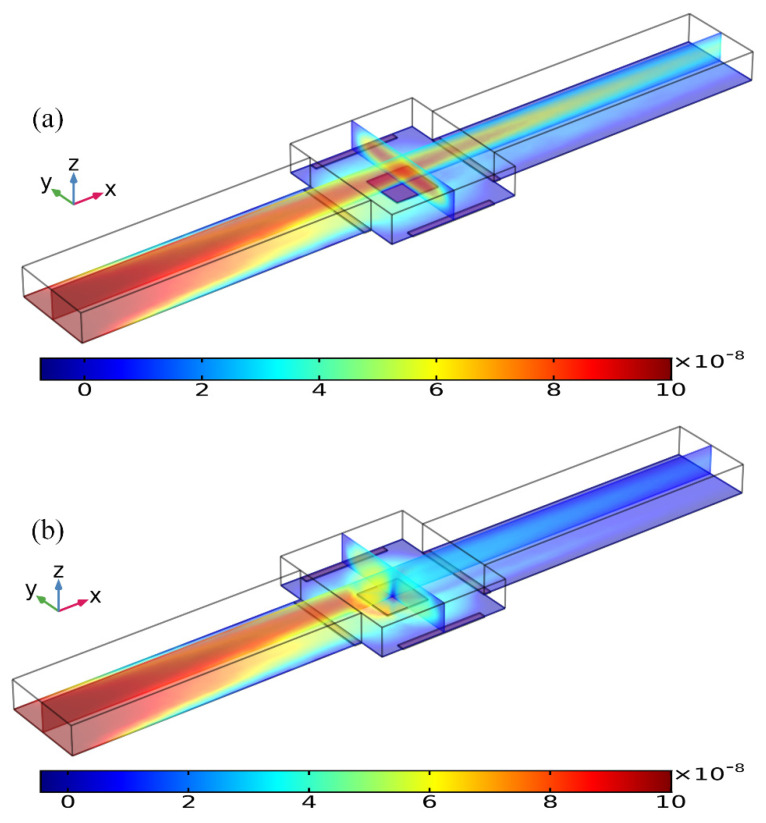
Concentration distribution of the incoming antigen injected from the inlet of a straight fluidic channel at *t* = 80 s, which is modified by the ROT-ICEO vortex flow field on the top of the square FE of 300 μm in edge length inside the central microchamber, when the peripheral DE array was excited by TW voltage signals of (**a**) *V*_0_ = 0 V, and (**b**) *V*_0_ = 8 V at *f* = 200 Hz with an average inlet flow rate of 50 um/s. (unit: mol/m^3^).

**Figure 11 micromachines-11-00739-f011:**
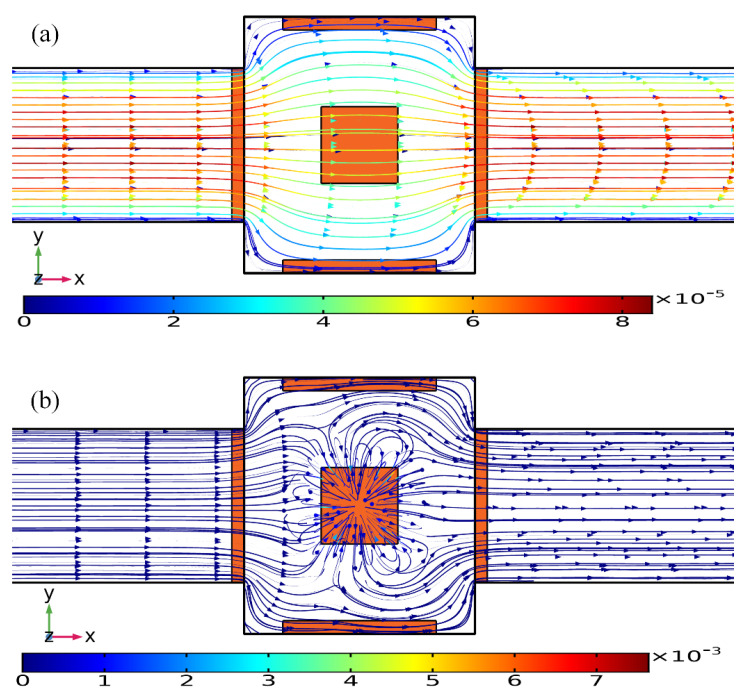
Influence of ROT-ICEO electroconvection on top of the central FE on the incoming laminar stream under the dynamic condition, when the driving voltage amplitude is (**a**) *V*_0_ = 0 V, and (**b**) *V*_0_ = 8 V at *f* = 200 Hz with an average inlet flow rate of 50 um/s (unit: m/s).

**Figure 12 micromachines-11-00739-f012:**
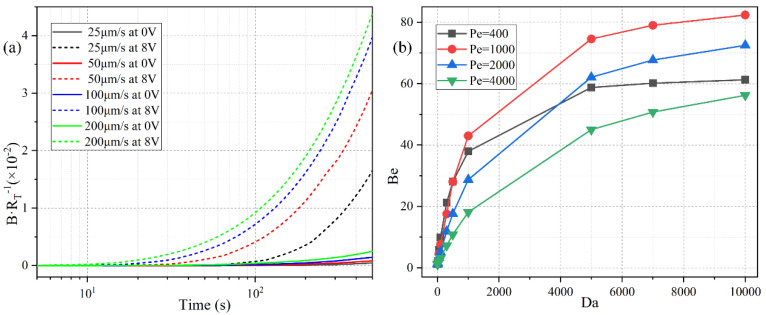
The quantitative study of the effect of ROT-ICEO on the degree of binding reaction improvement in the dynamic flow condition. (**a**) The normalized surface concentration of bound antigen with respect to the immobilized antibody under a voltage contrast between 0 V and 8 V for different inlet flow rate. (**b**) Influence of Da number on the binding enhancement factor *Be* after applying a rotating electric field of *V*_0_ = 8 V and *f* = 200 Hz at *t* = 300 s for distinct Peclet number. The simulation file for an inlet flow rate of 100 μm/s using Comsol Multiphysics 5.5 is included in the Appendix A.

**Table 1 micromachines-11-00739-t001:** AC voltage sequence for the three distinct power supply modes.

Power Supply Modes	1st Terminal	2nd Terminal	3rd Terminal	4th Terminal
(i)	*V*_1_ = *V*_0_cos(*ωt*)	*V*_2_ = *V*_0_cos(*ωt* + 90°)	*V*_3_ = *V*_0_cos(*ωt* + 180°)	*V*_4_ = *V*_0_cos(*ωt* + 270°)
(ii)	*V*_1_ = *V*_0_cos(*ωt*)	*V*_2_ = *V*_0_cos(*ωt*)	*V*_3_ = *V*_0_cos(*ωt* + 180°)	*V*_4_ = *V*_0_cos(*ωt* + 180°)
(iii)	*V*_1_ = *V*_0_cos(*ωt*)	*V*_2_ = *V*_0_cos(*ωt* + 180°)	*V*_3_ = *V*_0_cos(*ωt*)	*V*_4_ = *V*_0_cos(*ωt* + 180°)

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
