# Peer review of "A Numerical Investigation of Enhancing Microfluidic Heterogeneous Immunoassay on Bipolar Electrodes Driven by Induced-Charge Electroosmosis in Rotating Electric Fields"

_micromachines, 2020, doi:10.3390/mi11080739_

Round 1

Reviewer 1 Report

Authors have presented a numerical investigation to improve binding rate in microfluidic assays using induced charged electroosmosis in rotating electric fields. Parameters that can affect transport of fluid as well as binding molecules are analyzed using simulation. The manuscripts is well presented in adequately organized sections with appropriate/necessary figures and plots.

Author Response

Comment 1:

Authors have presented a numerical investigation to improve binding rate in microfluidic assays using induced charged electroosmosis in rotating electric fields. Parameters that can affect transport of fluid as well as binding molecules are analyzed using simulation. The manuscript is well presented in adequately organized sections with appropriate/necessary figures and plots.

Reply:

We thank you for the professional comments and insightful suggestions. Those comments are all valuable and very helpful for revising and improving our paper, as well as the important guiding significance to our researches. According to the comments of the referees, we carefully revised the manuscript to enable a better clarification of the originality of current work.

Reviewer 2 Report

The following comments may help the authors to improve the manuscript:

  1. The model looks at constant viscosity fluid, biological sample is however more complicated in term of viscosity and type of fluids. Please elaborate and perhaps provide a discussion in the manuscript. See for example, the fluid type: Newtonian and non-Newtonian: Microfluid Nanofluid 21, 37 (2017). https://doi.org/10.1007/s10404-017-1866-y
  1. Please provide the simulation file in the supplementary material.

Author Response

Comment 1:

  1. The model looks at constant viscosity fluid, biological sample is however more complicated in term of viscosity and type of fluids. Please elaborate and perhaps provide a discussion in the manuscript. See for example, the fluid type: Newtonian and non-Newtonian: Microfluid Nanofluid21, 37 (2017). https://doi.org/10.1007/s10404-017-1866-y

Reply:

Thank you for your professional and insightful comments.

It is well known that ICEO fluid motion vanishes in high conductivity buffer solutions due to both double layer shrinkage and ion overcrowding phenomenon inside the IDL. So, we prefer to study herein the effect of ROT-ICEO on improving microfluidic immunoassays in dilute electrolyte with electric conductivity usually no more than 0.02S/m. The aqueous electrolyte is a typical Newtonian fluid, and has a constant dynamic viscosity, serving as the most appropriate liquid medium for suspending the free antigens in the present analysis. On the other hand, recently, the necessity of manipulation of biofluids in small confinements has triggered a renewed interest in the dynamics of non-Newtonian fluid with a shear-rate-dependent viscosity. However, this is beyond the scope of current work with the purpose to provide a utilitarian reference for the selection of the parametric space in experiments with water-based Newtonian fluid. As a result, a uniform viscosity value η=0.001 Pa·s is used in the simulation to reconstruct the actual mechanical behavior of target antigens monodispersed in water solution. Please refer to Ref.[51-53] for a systematic knowledge of both Newtonian fluid and non-Newtonian fluid.

Please refer to the highlighted words in Page 6.

Comment 2:

  1. Please provide the simulation file in the supplementary material.

Reply:

We appreciate your useful comments.

Considering your suggestion, we include the simulation profile using Comsol Multiphysics 5.5 for an inlet flow rate of 100μm/s (Fig.12) in the Supplementary Material.

Please refer to the highlighted words in Page 17.
